# Peripheral Blood DNA Methylation Profiles Do Not Predict Endoscopic Post-Operative Recurrence in Crohn’s Disease Patients

**DOI:** 10.3390/ijms231810467

**Published:** 2022-09-09

**Authors:** Vincent W. Joustra, Andrew Y. F. Li Yim, Jessica R. de Bruyn, Marjolijn Duijvestein, Ishtu L. Hageman, Wouter J. de Jonge, Peter Henneman, Manon Wildenberg, Geert D’Haens

**Affiliations:** 1Department of Gastroenterology and Hepatology, Amsterdam Gastroenterology Endocrinology Metabolism, Amsterdam UMC, University of Amsterdam, 1105 AZ Amsterdam, The Netherlands; 2Genome Diagnostics Laboratory, Department of Clinical Genetics, Amsterdam UMC, University of Amsterdam, 1105 AZ Amsterdam, The Netherlands; 3Tytgat Institute for Liver and Intestinal Research, Amsterdam UMC, University of Amsterdam, 1105 BK Amsterdam, The Netherlands; 4Department of Gastroenterology and Hepatology, Radboud University Medical Centre, 6525 GA Nijmegen, The Netherlands

**Keywords:** epigenetics, DNA methylation, predictive biomarker, personalized medicine, endoscopic recurrence

## Abstract

Prediction of endoscopic post-operative recurrence (POR) in Crohn’s disease (CD) patients following ileocolonic resection (ICR) using clinical risk factors alone has thus far been inadequate. While peripheral blood leukocyte (PBL) DNA methylation has shown promise as a tool for predicting recurrence in cancer, no data in CD patients exists. Therefore, this study explored the association and predictive value of PBL DNA methylation in CD patients following ICR. From a cohort of 117 CD patients undergoing ICR, epigenome-wide PBL methylation profiles from 25 carefully selected patients presenting either clear endoscopic remission (*n* = 12) or severe recurrence (*n* = 13) were assessed using the Illumina MethylationEPIC (850K) array. No statistically significant differentially methylated positions (DMPs) or regions (DMRs) associated with endoscopic POR were identified (FDR *p* ≤ 0.05), further evidenced by the low accuracy (0.625) following elastic net classification analysis. Nonetheless, interrogating the most significant differences in methylation suggested POR-associated hypermethylation in the *MBNL1*, *RAB29* and *LEPR* genes, respectively, which are involved in intestinal fibrosis, inflammation and wound healing. Notably, we observed a higher estimated proportion of monocytes in endoscopic POR compared to remission. Altogether, we observed limited differences in the genome-wide DNA methylome among CD patients with and without endoscopic POR. We therefore conclude that PBL DNA methylation is not a feasible predictive tool in post-operative CD.

## 1. Introduction

Crohn’s disease (CD) is a chronic inflammatory bowel disorder (IBD), which is characterized by a relapsing and remitting course of inflammation that mostly affects the gastrointestinal (GI) tract [1].

Despite standard medical treatment, approximately 50% of patients require surgery within 10 years following diagnosis, which is rarely curative [2]. Approximately half of all patients undergoing ileocolonic resection (ICR) show endoscopic post-operative recurrence (POR) of disease within 6–12 months, the severity of which is assessed using the Rutgeerts score and has been shown to predict recurrence of clinical symptoms [3,4]. This group of patients might therefore benefit from early treatment intervention following surgery, ultimately lowering the incidence of endoscopic- and clinical disease recurrence. To this end, current clinical practice requires a risk assessment to determine if a patient is predisposed towards the development of endoscopic POR. Currently, these risk profiles are based on the presence of one or more clinical characteristics, such as active smoking, penetrating disease phenotype and previous IBD-related surgery [5,6,7]. Despite such efforts, a considerable number of CD patients still develop endoscopic POR. Therefore, supplementing these clinical characteristics with additional biomarkers would enable clinicians to intervene more accurately, thereby mitigating severe disease progression. However, to date, no validated biomarker can accurately predict endoscopic POR in CD patients [8].

Previous research, predominantly in cancer, has shown that the epigenome can be of clinical importance as a tool for predicting recurrence and subsequently providing adequate treatment choices to improve treatment responses [9,10,11,12]. Among the most widely studied epigenetic mechanisms is DNA methylation, which represents the covalent attachment of a methyl group to cytosine followed by a guanine (CpG). Aberrant cytosine methylation is generally associated with altered gene expression when it occurs in the promoter region, which is thought to occur through the inhibition of transcription factor binding [13]. In IBD, multiple studies have shown that differentially methylated loci across different cell types are relevant for chronic inflammatory disease states and are capable of distinguishing between phenotypes, underscoring their use as potential biomarkers [14,15,16,17,18,19]. While DNA methylation profiles are highly tissue-specific, the previous literature has shown that IBD-associated differences in methylation observed at the level of peripheral blood leukocytes (PBLs) can also be observed at the level of intestinal tissue, which is thought to result from continuous PBL–gut immune cell trafficking [17,20]. As PBLs are easily accessible and minimally invasive, their use in epigenetic biomarker research has gained increasing interest. In addition, epigenetic biomarkers, unlike genetic biomarkers, are capable of incorporating environmental and lifestyle factors—which might account for the missing heritability observed in IBD pathogenesis [21] as well as provide an avenue for interventional treatments [22].

To date, no study has sought to predict endoscopic POR in CD patients using epigenetic profiling. The identification of epigenetic biomarkers, measured at the time of the surgery, could potentially prevent POR and the need for repeated surgery. In this study, we therefore aimed to explore the association and predictive capabilities of DNA methylation from peripheral blood (PB) obtained immediately post-ileocolonic resection, with endoscopic POR in CD patients.

## 2. Results

### 2.1. Patient Characteristics and Clinical Parameters

A total of 25 CD patients with a median age of 31 (25–46) years old were included for analyses (Figure 1 and Table 1). Patients underwent clinical follow-up at 2, 6, 12 and 26 weeks, with endoscopic evaluation at week 26. During the endoscopic evaluation of mucosal disease, 12 patients were found to present with clear endoscopic remission (Rutgeerts score i0 or i1), whereas the remaining 13 patients presented with severe endoscopic POR (Rutgeerts score i3 or i4). The clinical parameters of patients with and without POR did not significantly differ for age (*p* = 0.51), smoking behavior (*p* = 0.14), penetrating disease phenotype (*p* = 0.54) or previous IBD-related resections (*p* = 0.15). By contrast, the endoscopic remission group consisted of 44% more females (*p* = 0.03), with a higher percentage of ileal disease (83.3% vs. 46.2%, *p* = 0.048) compared to the endoscopic POR group. No significant associations between active smoking (OR = 0.3; 95% CI 0.04–1.69; *p* = 0.16), previous IBD-related surgery (OR = 4.8; 95% CI 0.46–51.87; *p* = 0.19) or penetrating phenotype (OR = 1.88; 95% CI 0.25–14.08; *p* = 0.54) and endoscopic POR were observed.

### 2.2. Differential Methylation Analysis

We initially interrogated the data through explorative approaches. Principal component analysis (PCA) did not show a clear separation between patients in endoscopic remission (orange) versus patients with endoscopic POR (blue; Figure 2), indicating that a predisposition to endoscopic POR does not visibly manifest in a genome-wide fashion.

We subsequently performed comparative analyses by comparing patients that went on to develop endoscopic POR with those that remained in remission. Following correction for multiple testing (FDR *p* ≤ 0.05), we were not able to identify any probes that presented significant differential methylation. Interrogating the 20 most significant differences in methylation, we found them to be annotated to 15 known genes (Table 2).

The majority of these CpGs (18 out of 20) displayed hypermethylation in patients developing endoscopic POR compared to remission. The two probes showing hypomethylation in endoscopic POR versus remission were annotated to *SLC43A2* and *GLI3*, respectively. Among the top 20, eight CpGs in particular presented a large (>10%) mean difference in methylation, of which only four were annotated to genes: namely, cg16537483 (*MBNL1*), cg22120095 (*CACNA2D2*), cg26418147 (*RAB29*), and cg03050981 (*LEPR*; Figure 3).

In an effort to properly assess the prognostic capability of the DNA methylome in discriminating endoscopic remission from endoscopic POR, we performed an elastic net binomial classification analysis. Through repeated cross-validation, we identified a set of 27 CpGs as being the most capable of discriminating patients developing endoscopic POR from patients remaining in remission based on the training set. However, testing this model on the withheld test set yielded an accuracy of 0.625 (sensitivity = 0.6; specificity = 0.67; Figure 4). Taken together, we observed that individual loci by themselves offer a limited capability of distinguishing patients that will develop endoscopic POR from patients that will remain in remission.

As peripheral blood contains a heterogeneous cell population, we estimated the cellular composition based on DNA methylation data [23]. Comparing the estimated cellular proportions between patients that went on to develop endoscopic POR with patients that would remain in remission suggested a significant difference in monocyte proportions (*p*-value = 0.008). Specifically, patients that went on to develop endoscopic POR presented a higher monocyte population compared to patients that remained in remission (Figure 5).

Importantly, we found no clear evidence that the proportion of monocytes among the patients developing endoscopic POR was associated with gender, smoker status, prior IBD-related surgery, penetrating disease phenotype or high-dose vitamin D (Appendix A). 

## 3. Discussion

This study aimed to explore the association and predictive capability of peripheral blood DNA methylation with endoscopic POR in 25 well-characterized adult CD patients following ileocolonic resection. To do so, we measured genome-wide DNA methylation profiles in peripheral blood DNA samples taken as soon as oral intake was resumed following surgery. Extensive comparative analyses, performed through linear regression and classification analyses, were unable to identify a clear signal capable of discerning patients that would eventually develop endoscopic POR from patients that would remain in endoscopic remission. Our results suggest that peripheral blood DNA methylation in patients immediately post-ICR does not differ significantly in patients that ultimately develop endoscopic recurrence compared to patients that show clear endoscopic remission after 26 weeks of follow-up.

Nonetheless, we identified cg16537483 (*MBNL1*), cg26418147 (*RAB29*) and cg03050981 (*LEPR*) as promising candidates for further research based on previous reports. The most prominent difference was found for cg16537483, which occurs in the gene body of *MBNL1* (Muscleblind Like Splicing Regulator 1). *MBNL1* encodes an RNA-splicing protein with a well-characterized role in the pathogenesis of myotonic dystrophy [24,25,26,27] that was recently associated with MLL-rearranged leukemia [28] and colon cancer [29]. Interestingly, *MBNL1* deficiency in mice has been shown to impair the differentiation of fibroblasts into myofibroblasts as part of the fibrotic response, reduces wound healing [30] and alters T-cell receptor signaling [31]. Furthermore, *MBNL1* has been implicated as a negative regulator of TGF-β in endocardial cells [32,33]. In postoperative CD, high expression of TGFB1 in the uninflamed intestinal mucosa at the time of surgery has been associated with higher cumulative clinical recurrence rates after 26 weeks of follow-up (20% vs. 0%, *p* = 0.02) [34,35]. Taken together, these observations suggest a potential role of MBNL1 in the postoperative CD course by altering the fibrotic response, mucosal wound healing and T-cell activation—all previously implicated in the etiology of POR [34,36]. However, further research addressing the downstream functionality of altered *MBNL1* methylation is needed.

The second probe of interest was cg26418147 and was found near *RAB29* (Member RAS Oncogene Family), which encodes a small GTPase Rab. *RAB29* is a key regulator in vesicle trafficking and is essential for maintaining the integrity of the endosome–trans-Golgi network structure. Notably, *RAB29* activates *Lrrk2* [37,38,39,40]—a gene that has been identified by genome-wide association studies (GWAS) as a major susceptibility gene for CD [41]. Loss of *Lrrk2* in mice results in increased inflammation and IgA expression upon DSS-treatment [42,43]. By contrast, overexpression of *Lrrk2* results in an increased severity of DSS-induced colitis in mice [44]. These studies highlight that while the exact role of *Lrrk2* in gut inflammation is still unclear [45,46], altering the expression of *Lrrk2* through RAB29 might negatively affect inflammation.

Lastly, we identified cg03050981 near *LEPR* (*LEP-R*), which functions as a receptor for leptin and is therefore mostly produced in adipocytes. Notably, leptin production has also been reported in the intestinal epithelial cells, macrophages, T-cells and NK-cells—predominantly during episodes of acute inflammation [47]. In addition, leptin-signaling has been shown to regulate the inflammatory response through cytokine production, activation of monocyte/macrophages and wound healing [48,49,50].

The effect of leptin signaling on wound healing has been demonstrated by multiple rodent models, showing lower intestinal injury scores, increased wound healing, decreased time till complete mucosal healing and a higher cell proliferation when treated with leptin following injury [51,52]. By contrast, in DSS-induced colitis of *LEPR*^−/−^ mice, increased weight loss and mucosal necrosis, deep colonic epithelial damage and loss of epithelial cells was observed compared to control mice [53]. Our data suggests that altered methylation in *LEPR* is potentially involved in the etiology of post-operative endoscopic recurrence of CD.

To the best of our knowledge, this is the first study that has explored the association and predictive capability of DNA methylation signatures measured directly following surgery with the occurrence of endoscopic POR at week 26. We believe the main strength of this study lies in the strict selection of our cohort, which provides high confidence in the observed phenotype (endoscopic recurrence or remission). The collection of samples and strict follow-up were all performed according to a pre-specified protocol [54]. In addition, to minimize any inter-observer variation in endoscopic scoring, all endoscopies were videotaped and scored through central reading by two expert endoscopists (G.D. and P.B.).

There are several limitations to address in this study. First of all, the case–control EWAS design of this study limits causal inference. Secondly, we performed analyses on a limited sample size of 25 CD patients. However, the purpose of this study was explorative in nature and no reference for a sample size calculation was available. In order to increase our confidence in finding significant differences, we therefore used samples from patients with a very strict characterization of endoscopic recurrence (centrally read endoscopies at a fixed time point of 26 weeks). Despite adhering to these strict classification criteria, with high confidence in endoscopic scoring through central reading, limited differences in DNA methylation were observed between both groups.

Based on our data, a similar case–control design adhering to the current sample size estimations [55] would require too large a sample size in order to detect significant differences between groups with adequate power.

Thirdly, while IBD-associated differences in PBL DNA methylation have previously been observed in intestinal tissue as well [17,20], it has not been said that such an observation is unequivocally true. This is especially important given the fact that the mucosal tissue is composed of a wide variety of cells not present in PBLs—each of which presents its own DNA methylome.

Lastly, our predictive model cannot be generalized to predict borderline endoscopic recurrence in CD patients due to the exclusion of these patients (Rutgeerts i2a and i2b), which encompasses the majority of the scored endoscopies (83 out of 117).

Future experiments addressing the downstream functionality and biology of the identified DMP-associated genes of interest (*MBNL1*, *RAB29* and *LEPR*) are needed to understand their potential etiological role in early endoscopic POR. In addition, our data suggests that the potential for PBL DNA methylation to be used as a stand-alone biomarker platform in post-operative CD is limited. It may therefore be more feasible to turn our attention to other -omics to improve classification of CD patients in the post-operative setting.

The link between the gut microbiome and the epigenome has been shown in multiple studies and is suggested to serve as a key tool for identifying targets for diagnosis and treatment [56].

Recent evidence suggests that both luminal- and mucosa-adherent microbiota are associated with the occurrence of endoscopic POR [57,58]. Machiels et al. has shown that mucosa-adherent microbiota at baseline have a moderate–good discriminative power to predict endoscopic POR (AUC 0.738) and outperformed clinical parameters alone (AUC 0.612). Interestingly, combining the microbiota and clinical data slightly improved the performance of their model (AUC 0.779). We therefore hypothesize that future studies combining clinical-, microbiome- and epigenetic data derived from intestinal tissue might provide an even more accurate classification of which patients to treat early following resection.

Altogether, we observed limited differences and predictive capabilities with genome-wide DNA methylation profiles measured at surgery associated with endoscopic POR at week 26 in 25 well-characterized CD patients. We therefore conclude that the PBL DNA methylome *alone* cannot discriminate CD patients with endoscopic POR from remission at the time of surgery. A combination of multi-omics, including clinical parameters, to predict endoscopic POR might be more suitable. 

## 4. Materials and Methods

### 4.1. Patient Cohort Selection

Peripheral blood samples were obtained from established CD patients after undergoing ICR at 17 regional and academic hospitals across the Netherlands and Belgium between February 2014 and June 2017. Patients participated in the DETECT trial [54]. In short, the goal of this randomized controlled trial was to determine the efficacy of high-dose vitamin D as an anti-inflammatory treatment in patients following primary ileocolonic resection. From a total cohort of 117 patients with a follow-up endoscopy at week 26, twenty-five patients were selected with either clear endoscopic remission (modified Rutgeerts i0 or i1) or severe endoscopic POR (modified Rutgeerts i3 or i4). We purposely removed patients with a modified Rutgeerts score of i2a or i2b due to ongoing debates regarding the classification of these stages as well as high rates of inter-observer variability for this score [59,60,61]. A detailed description of the modified Rutgeerts scoring can be found in Appendix A [62].

All ileocolonoscopies were videotaped, after which two experienced IBD endoscopists (G.D. and P.B.) performed independent and blinded scoring. Any disagreements were resolved through an adjudication meeting between the two readers. The study was conducted according to the guidelines of the Declaration of Helsinki, and the assembly of this cohort was approved by the medical ethics committee of the Academic Medical Center (METC W19_212# 19.257, 3 June 2019). None of the eligible patients objected against the reuse of their peripheral blood DNA for the purpose of this study.

A single peripheral blood sample for each patient was collected directly following surgery prior to the randomization and administration of either placebo or high-dose vitamin D, as part of the DETECT study protocol [54]. This peripheral blood sample was collected in a 4.5 mL EDTA tube using standard venipuncture and stored at −80° Celsius until further handling. A flow-chart of the patient selection is shown in Figure 1 and a summary of important baseline characteristics is shown in Table 1.

### 4.2. DNA Isolation, Quality Control and Bisulfite Conversion

Genomic DNA from peripheral blood was isolated with the QIAamp DNA blood mini kit according to the manufacturer’s protocol.

The resulting quality was then assessed on a 1% agarose gel to ensure a high molecular weight, whereupon the concentration of dsDNA was measured using a FLUOstar. The genomic DNA was then made equimolar and bisulfite-treated using a Zymo EZ DNA Methylation Kit, after which the whole-genome DNA methylation profiles were quantified using an Illumina HumanMethylation EPIC BeadChip array, at the Core Facility Genomics, Amsterdam UMC, Amsterdam, The Netherlands.

### 4.3. Clinical Analysis

We used descriptive statistics to assess the baseline characteristics of the 25 selected patients. Continuous data are expressed as median (IQR), and categorical data as frequencies and percentages. Differences in distribution between patients in endoscopic remission or recurrence were assessed using a chi-square test (categorical variables) or Mann–Whitney U test (continuous variables). Two-tailed probabilities were usedm with a *p*-value of ≤0.05 considered as statistically significant. Univariate binary logistic regression analyses were used to explore the association of active smoking, penetrating disease phenotype (Montreal B3) and previous IBD-related resections with the occurrence of endoscopic POR at week 26. Analyses of clinical data were performed in IBM SPSS statistics version 26.

### 4.4. Methylation Analysis

Raw DNA methylation data were imported into R [63] (version 4.0.5, Vienna, Austria) using the Bioconductor package *minfi* [64] (version 1.36), after which functional normalization [65] was used. Quality control was conducted using *MethylAid* [66] (version 1.24). For the remainder of our analysis, we excluded all probes hybridizing to allosomes due to the mixed-sex cohort. Probes hybridizing with known genetic variants in either the probe binding site or in the CpG of interest (minor allele frequency > 0.01) were removed. M-values (M  =  log2(M/U)) were utilized for the statistical analyses, whereas β-values (β  =  M/(M  +  U  +  100)) were used for visualization [67]. Differentially methylated positions (DMPs) and regions (DMRs) were identified through generalized linear regression analyses using *limma* [68,69] (version 3.46) and *DMRcate* [70] (version 2.4.1), respectively. For the generalized linear models, we adjusted for age and sex. Loci were considered statistically significant if their Benjamini-Hochberg (BH) adjusted *p*-value was <0.05. Elastic net classification analysis was performed using *glmnet* [71] (version 4.1.1). In short, the total dataset was split into a training (60%) and test (40%) set, whereupon 50 repetitions of 5-fold cross-validation were employed to train the model. Final validations were performed on the withheld test set. The cellular composition was estimated using the estimateCellCounts function in minfi using the IDOL dataset [23] as reference, whereupon comparative analyses were completed using two sample *t*-tests implemented in *genefilter* [72] (version 1.72.1). Visualizations were put together in ggplot2 [73] (version 3.3.2).

## Figures and Tables

**Figure 1 ijms-23-10467-f001:**
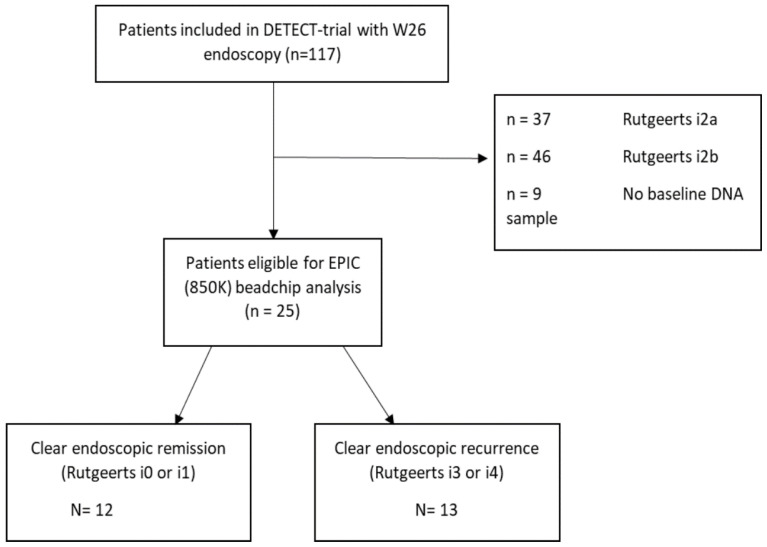
Patient selection. Flowchart demonstrating the selection of patients.

**Figure 2 ijms-23-10467-f002:**
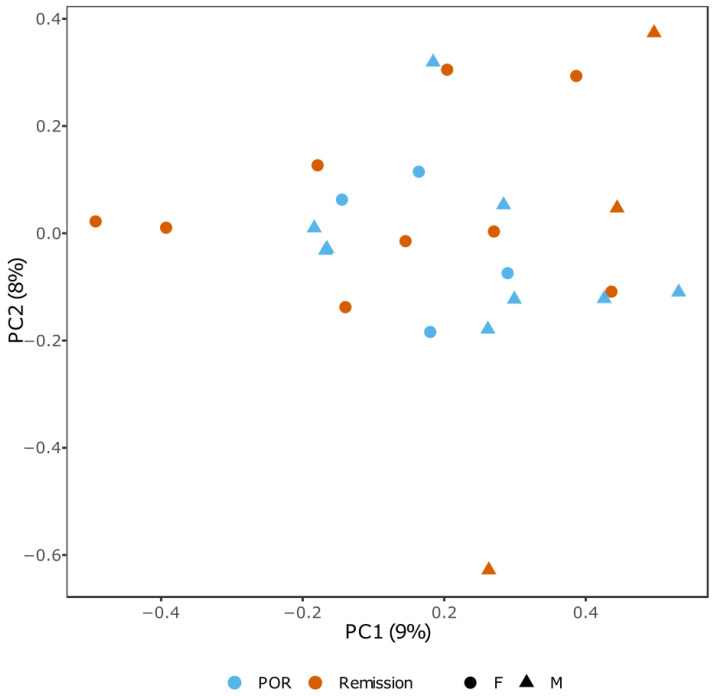
PCA plot of endoscopic remission vs. recurrence. Principal component analysis of genome-wide DNA methylation profiles in patients with endoscopic POR (blue) or remission (orange) at week 26, stratified for gender.

**Figure 3 ijms-23-10467-f003:**
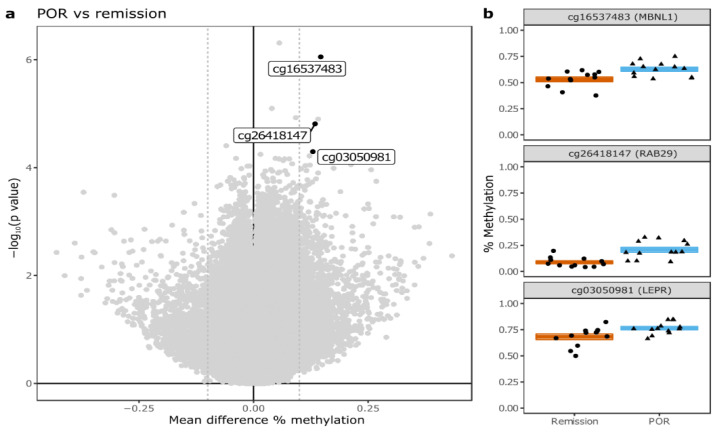
Differential methylation endoscopic recurrence vs. endoscopic remission of *LEPR*, *MBNL1*, and *RAB29* (**a**). Volcano plot summarizing the difference in whole-genome DNA methylation between patients with endoscopic POR vs. remission at week 26. Each grey dot represents one of the 850,000 CpGs interrogated. The black dots represent CpGs of interest discussed in this manuscript. The x-axis shows the mean difference in methylation percentage, the Y-axis shows the magnitude of the p-value for that particular CpG (**b**). Boxplots of DMP-associated genes of interest cg03050981 (*LEPR*), cg16537483 (*MBNL1*) and cg26418147 (*RAB29*) in (**b**).

**Figure 4 ijms-23-10467-f004:**
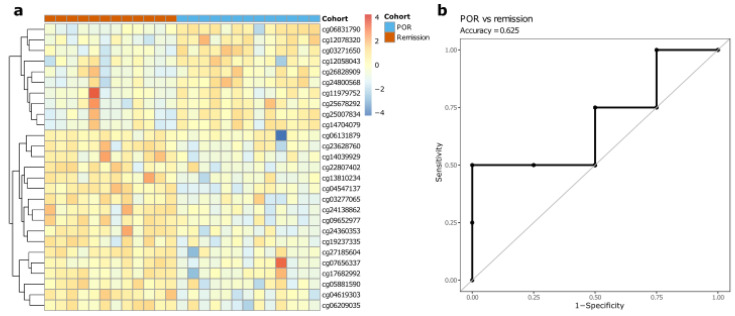
Elastic net classification analysis endoscopic recurrence vs. remission. (**a**): Training an elastic net classification model to distinguish patients developing POR from patients that remain in remission identified 27 CpG-probes. (**b**): Receiver operating characteristic curve showing an accuracy of 0.625 on the test set.

**Figure 5 ijms-23-10467-f005:**
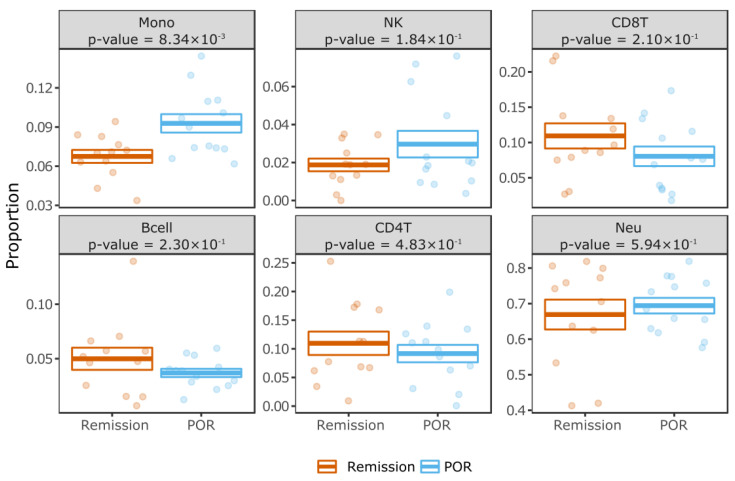
Estimated cell distributions endoscopic recurrence vs. remission. Estimated blood cell distribution of the monocytes, NK-cells, CD8+ T-cells, B-cells, CD4+ T-cells and neutrophils using IDOL dataset. The x-axis of each box indicates the difference between patients the remain in endoscopic remission (orange) vs POR (blue) with corresponding p-values as calculated using two-sample t-tests shown above. The y-axis of each box shows the proportion of that particular cell type. A significantly higher proportion of monocytes at baseline were observed for patients ultimately developing POR.

**Table 1 ijms-23-10467-t001:** Baseline characteristics.

	Endoscopic Remission (*n* = 12)	Endoscopic Recurrence (*n* = 13)	*p*-Value
Female, *n* (%)	9 (75)	4 (30.8)	**0.03**
Age, years, median (IQR)	30 (21–42)	31 (25–53)	0.51
Ethnic background, *n* (%) - Caucasian	10 (83.3)	10 (76.9)	0.72
C-reactive protein, mg/L, median (IQR)	25.5 (3–84.5)	62.6 (13.6–104)	0.28
Faecal calprotectin, ug/g, median (IQR)	454.5 (111–1650.5)	805 (303.8–1602.5)	0.39
Baseline CDAI score, median (IQR)	104.5 (62.3–161.3)	144 (101–292)	0.11
Disease location, *n* (%) - Ileal disease (L1) - Ileocolonic disease (L3)	10 (83.3) 2 (16.7)	6 (46.2) 7 (53.8)	**0.048**
Disease behavior, *n* (%) - Non stricturing/penetrating (B1) - Stricturing (B2) - Penetrating (B3) - Perianal disease (p)	3 (25) 7 (58.3) 2 (16.7) -	3 (27.3) 5 (45.5) 3 (27.3) 2 (15.4)	0.78
Previous IBD related surgery, *n* (%)	1 (8.3)	4 (30.8)	0.15
Previous medical treatment, *n* (%) - Immunomodulator (AZA, 6MP, MTX) - Anti-TNF (IFX and/or ADA)	7 (58.3) 5 (41.7)	7 (58.3) 7 (58.3)	0.82 0.54
Smoking, *n* (%) - Active - Non-smoker	5 (41.7) 7 (58.3)	2 (15.4) 11 (84.6)	0.14

*p* values < 0.05 are marked in bold and indicate statistical significance. IQR, interquartile range; CDAI, Crohn’s disease activity index; AZA, azathioprine; 6MP, 6-mercaptopurine; MTX, methotrexate; IFX, infliximab; ADA, adalimumab.

**Table 2 ijms-23-10467-t002:** Top 20 DMPs at baseline, comparing severe endoscopic recurrence vs. endoscopic remission (assessed at week 26).

CpG ID	*p* Value	adj. *p* Value	Beta	Annotated Genes
cg22681074	4.89 × 10^−7^	0.375	0.056	*GJC2*
cg16537483	8.85 × 10^−7^	0.375	0.147	*MBNL1*
cg15599437	7.97 × 10^−6^	0.966	0.040	
cg20677058	1.18 × 10^−5^	0.966	0.092	*AKR7L*
cg22120095	1.26 × 10^−5^	0.966	0.141	*CACNA2D2*
cg26418147	1.54 × 10^−5^	0.966	0.134	*RAB29*
cg25215028	2.06 × 10^−5^	0.966	0.108	
cg05128623	3.92 × 10^−5^	0.966	−0.060	*SLC43A2*
cg03050981	5.05 × 10^−5^	0.966	0.129	*LEPR*
cg12919469	5.68 × 10^−5^	0.966	0.015	*TMC4*
cg07528209	6.05 × 10^−5^	0.966	0.122	
cg01094108	6.75 × 10^−5^	0.966	−0.038	*GLI3*
cg11551901	6.76 × 10^−5^	0.966	0.048	*SEC31B*
cg22995183	6.87 × 10^−5^	0.966	0.078	*MRTFB*
cg01543603	6.90 × 10^−5^	0.966	0.056	*ANKRD11*
cg08514511	6.91 × 10^−5^	0.966	0.068	*FRK*
cg14574579	7.37 × 10^−5^	0.966	0.064	*UMODL1*
cg05725940	8.50 × 10^−5^	0.966	0.084	*GSDMB*
cg00871238	8.81 × 10^−5^	0.966	0.212	
cg14219900	8.96 × 10^−5^	0.966	0.123	

## Data Availability

The raw DNA methylation data generated in this study has been published under controlled access for research purposes at the European Genome-phenome Archive at EGAS00001006121. All R scripts have been made available on GitHub and can be found at https://gitlab.com/ND91/hgprj0000032_cdpbpor.

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
