# Peer review of "Peripheral Blood DNA Methylation Profiles Do Not Predict Endoscopic Post-Operative Recurrence in Crohn’s Disease Patients"

_ijms, 2022, doi:10.3390/ijms231810467_

Round 1
Reviewer 1 Report
Attached please find my comments.

Author Response
Dear Reviewer,
We are grateful for the time and effort taken in critically reading our manuscript. We hope that we have responded adequately to the comments in our point-by-point response below and that the revised manuscript (attached) is now suitable for publication in International Journal of Molecular Sciences.
Kind Regards,
Vincent Joustra, MD
Andrew Li Yim, PhD
------------------------------------------------------------------------------------
Reviewer 1:
In this paper, Vincent and co-authors have conducted a cohort study on 25 Crohn’s disease (CD) patients. They reported that peripheral blood leukocyte (PBL) DNA methylation could not be used as a biomarker to predict the recurrence of post-operative CD. This cohort study is very significant and meaningful, and provides evidences to the controversy in the literature and clinical diagnosis regarding this association. However, some detail information is missing which is crucial for interpreting the results (see below). In addition, I have a few suggestions and comments which I think should be addressed before being published.
- The authors failed to address the criteria of cohort selection. How were these 25 patients selected from 117 patients?
We would like to thank the author for addressing this item. Our study included a post-hoc analysis on a subset of patients as included in the DETECT trial1, a randomized controlled trial that evaluated the efficacy of high-dose vitamin D as anti-inflammatory treatment in patients following primary ileocolonic resection. These patients were recruited between February 2014 and June 2017 in 17 regional and academic hospitals across the Netherlands and Belgium. The primary outcome for which our study aimed to explore the potential of peripheral blood DNA methylation as predictive biomarker, was endoscopic POR at week 26. Hence the studied patients were selected specifically based on this outcome.
We specifically chose to select patients with clear remission (Rutgeerts i0 or i1, thus mostly absent mucosal inflammation) and patients with clear recurrence (Rutgeerts i3 or i4, thus diffuse inflammation, large ulcers and/or stenosis) at week 26 to explore the capability of DNA methylation in such ‘extreme’ outcomes. Due to ongoing debate on the position of Rutgeerts scores i2a and i2b2, 3 in recurrence or remission, relatively high rates of inter-observer variability for these scores4, as well as the exploratory nature of this study, we chose not to include all 117 patients. While this was described in Figure 1, we have now included this in detail in the materials and methods section ‘patient cohort selection’ lines 272-280 and provided a detailed description of the modified Rutgeerts’ score in table S1.
- Some critical information of patient characteristics was not provided, for example: patient disease history, recent prescription, living environment, lifestyle etc. These personal characteristics all contribute to the disease development and should be taken into consideration when do the association analysis.
We agree with the reviewer that several clinical- or environmental factors could potentially influence disease development and the DNA methylome. The most well-known factors that could have such a potential effect are age, gender, smoking behavior and cell distribution (in heterogeneous cell samples such as peripheral blood). We therefore collected this information for each of the included patients (see table 1). Furthermore, we explored the possibility of a difference in cell population affecting our analysis in figure 5. While the living environment (rural or urban) and other lifestyle factors (such as diet) could affect DNA methylation analyses, we unfortunately do not possess this information for the included patients.
Besides factors that could influence the DNA methylation of included patients, we agree that it is paramount to clinically characterize included patients in detail. To that end, we included ‘disease location’ and ‘disease behavior’ according to the Montreal classification5 as well as previous surgeries and previous medical treatment prior to surgery in table 1. Besides the higher percentage of female and ileal diseased patients in the endoscopic remission group, all other factors were well balanced in our cohort. Both gender and disease location have not been recognized as risk factors for endoscopic POR in current guidelines as well as our own analyses6-9.
As our selection of patients were subsequently treated with placebo or high-dose vitamin D in the post-operative setting no further prescriptions were administered. In addition, the DETECT trial did not show a significant effect of vitamin D treatment on the occurrence of endoscopic POR at week 26 and the peripheral blood samples which were analyses in this study were taken before administration of vitamin D1. Therefore we do not believe the different study arms in the DETECT trial could have affected our current analysis.
- Page 10 Line 214 claimed “there was no reference for a sample size calculation”, however, the patient selection criteria in current study is biased: 1) all selected patients were female; 2) gender difference was not considered, as all selected patients were around age 20-50. The comparison from current study was not convincing due to the limited patient size and unspecified patients selection criteria in each group.
We acknowledge that one of the factors limiting clinical interpretation of our results is the small patient size of our selected cohort, which accordingly posits the current study as an exploratory design. While this sample size could have been larger, we hypothesized that the inclusion of patients scored as Rutgeerts i2a or i2b would provide noise in distinguishing clear remission from recurrence, as indicated in point 1.
We disagree however that all selected patients were female (see table 1: in total 13 out of 25 (52%) were female). Nonetheless, the number of patients presenting remission were enriched for females, which we accordingly included as a covariate in our analyses (lines 326-327). That being said, current guidelines do not indicate any clear association of gender with an increased risk of developing endoscopic POR6-8.
We are not entirely sure what the issue is with a population aged between 20 and 50 years old given that this aptly represents the population of patients presenting with POR10. We observed no significant difference in the distribution of age between the recurrence or remission groups (median 31 (IQR 25-53) vs median 30 (IQR 21-42), p =0.51), indicating that there is no bias between the cases and controls. Nevertheless, our model did account for age as stated in the methods (lines 326-327).
- The peripheral blood sample collection method was not clear. Please rephrase “Peripheral blood samples were taken as soon as post-operative oral intake resumed prior to the randomization and administration of either placebo or high-dose vitamin D” on page 11 line 276
We thank the reviewer for pointing out the need to clarify this section. The peripheral blood samples were collected using standard venipuncture in a 4.5mL EDTA tube. Samples were collected as soon as possible following surgery and prior to randomization into the high-dose vitamin D group or placebo group. We rephrased this part accordingly in the material and methods, section “patient cohort selection”.
References:
- de Bruyn JR, Bossuyt P, Ferrante M, et al. High-Dose Vitamin D Does Not Prevent Postoperative Recurrence of Crohn's Disease in a Randomized Placebo-Controlled Trial. Clin Gastroenterol Hepatol 2020.
- Riviere P, Vermeire S, Irles-Depe M, et al. No Change in Determining Crohn's Disease Recurrence or Need for Endoscopic or Surgical Intervention With Modification of the Rutgeerts' Scoring System. Clinical Gastroenterology and Hepatology 2019;17:1643-1645.
- Bachour SP, Shah RS, Lyu RS, et al. Mild neoterminal ileal post-operative recurrence of Crohn's disease conveys higher risk for severe endoscopic disease progression than isolated anastomotic lesions. Alimentary Pharmacology & Therapeutics 2022;55:1139-1150.
- Ma C, Gecse KB, Duijvestein M, et al. Reliability of Endoscopic Evaluation of Postoperative Recurrent Crohn's Disease. Clinical Gastroenterology and Hepatology 2020;18:2139-+.
- Satsangi J, Silverberg MS, Vermeire S, et al. The Montreal classification of inflammatory bowel disease: controversies, consensus, and implications. Gut 2006;55:749-53.
- Lamb CA, Kennedy NA, Raine T, et al. British Society of Gastroenterology consensus guidelines on the management of inflammatory bowel disease in adults. Gut 2019;68:s1-s106.
- Gionchetti P, Dignass A, Danese S, et al. 3rd European Evidence-based Consensus on the Diagnosis and Management of Crohn's Disease 2016: Part 2: Surgical Management and Special Situations. J Crohns Colitis 2017;11:135-149.
- Nguyen GC, Loftus EV, Jr., Hirano I, et al. American Gastroenterological Association Institute Guideline on the Management of Crohn's Disease After Surgical Resection. Gastroenterology 2017;152:271-275.
- Joustra V, Duijvestein M, Mookhoek A, et al. Natural History and Risk Stratification of Recurrent Crohn's Disease After Ileocolonic Resection: A Multicenter Retrospective Cohort Study. Inflamm Bowel Dis 2021.
- Torres J, Mehandru S, Colombel JF, et al. Crohn's disease. Lancet 2017;389:1741-1755.

Reviewer 2 Report
Predicting recurrence of Crohn’s disease (CD) in patients whounderwent ileocolonic resection (ICR) has been inadequate also needing clinical risk factors. The authors challenged the use of clinical risk factors measured as the Rutgeerts score’s by adopting peripheral blood leukocyte (PBL) DNA methylation as a predictive clinical marker of recurrence, to predicting endoscopic post-operative recurrence (POR). They evaluated in a small sample of 25 CD patients in a double blind, clinical trial and found no conclusive correlations with endoscopy at week 26 of ICR. To achieve a difference, a much larger sample size is needed in concert with carefully designed clinical trials. The authors concluded that peripheral blood leukocyte (PBL) DNA methylation provided a feasible direction for future studies.
The abstract was informative in summarizing the research findings. The Introduction gave a detailed background of many aspects of the inadequacy of present biomarkers, and the nature of the clinically unmet needs.
The Methods were very well designed. The section gave detailed descriptions of patient cohort selection by a Modified Rutgeerts’ score, DNA isolation, clinical analyses, and methylation analyses. The statistical analyses were very sophisticatedly well-performed.
The results were clearly described, and the data were convincingly presented. The authors presented a detailed discussion in which they stated that no conclusive study results were provided by the study. They suggested that a study based on evaluation of three identified closely related 3/20 DMPs cg16537483 (MBNL1), cg26418147 (RAB29) and cg03050981 (LEPR) epigenetic markers was a feasible future direction. The authors also suggested a role of MBNL1 in the postoperative CD course by altering the fibrotic response, mucosal wound healing and T-cell activation. These markers have all been implicated in the etiology of POR.
The Conclusions were factual and appropriate. The authors asserted that a similar case-control design adhering to the current sample size estimations would require too large a sample size in order to detect significant differences between groups with adequate power. Nonetheless, the authors promised further downstream research.

Author Response
Dear Reviewer,
We are grateful for the time and effort taken in critically reading our manuscript and subsequent consideration for publication. We have made several adjustments based on the comments provided by reviewer 1 and hope that the revised manuscript (attached) is now suitable for publication in International Journal of Molecular Sciences.
Kind Regards,
Vincent Joustra, MD
Andrew Li Yim, PhD

Round 2
Reviewer 1 Report
I think the manuscript revision has significantly improved, and my concerns and questions have been addressed.